# Maintenance proton pump inhibitor use and risk of colorectal cancer: a Swedish retrospective cohort study

Qing Liu ,[1] Xinchen Wang,[2] Lars Engstrand,[1] Omid Sadr-Azodi,[3,4] Katja Fall,[5,6] Nele Brusselaers [1,7,8]

**Correspondence to**
Qing Liu;
qing.liu2017@outlook.com

## ABSTRACT

**Objectives** We aimed to evaluate the risk of colorectal adenocarcinoma (CRA) associated with long-term use of proton pump inhibitors (PPIs) in a large nationwide cohort.

**Design** Retrospective cohort study.

**Setting** This research was conducted at the national level, encompassing the entire population of Sweden.

**Participants** This study utilised Swedish national registries to identify all adults who had ≥180 days of cumulative PPI use between July 2005 and December 2012, excluding participants who were followed up for less than 1 year. A total of 754 118 maintenance PPI users were included, with a maximum follow-up of 7.5 years.

**Interventions** Maintenance PPI use (cumulative≥180 days), with a comparator of maintenance histamine-2 receptor antagonist ($H_2$RA) use.

**Primary and secondary outcome measures** The primary outcome measure was the risk of CRA, presented as standardised incidence ratios (SIRs) with 95% confidence intervals (CIs). Subgroup analyses were performed to explore the impact of indications, tumour locations, tumour stages and the duration of follow-up. A multivariable Poisson regression model was fitted to estimate the incidence rate ratios (IRRs) and 95% CIs of PPI versus $H_2$RA use.

**Results** Maintenance PPI users exhibited a slightly elevated risk of CRA compared to the general population (SIR 1.10, 95% CI=1.06 to 1.13) for both men and women. Individuals aged 18–39 (SIR 2.79, 95% CI=1.62 to 4.47) and 40–49 (SIR 2.02, 95% CI=1.65 to 2.45) had significantly higher risks than the general population. Right-sided CRA showed a higher risk compared to the general population (SIR 1.26, 95% CI=1.20 to 1.32). There was no significant difference in the risk of CRA between maintenance PPI users and maintenance $H_2$RA users (IRR 1.05, 95% CI=0.87 to 1.27, *p*<0.05).

**Conclusions** Maintenance PPI use may be associated with an increased risk of CRA, but a prolonged observation time is needed.

## INTRODUCTION

Proton pump inhibitors (PPIs) are one of the most commonly prescribed drugs globally and are widely used for the treatment and prevention of various gastric acid-related disorders.[1 2] The use of both prescription and over-the-counter PPIs has increased in

## STRENGTHS AND LIMITATIONS OF THIS STUDY

⇒ High number of enrolled individuals and person-years.
⇒ High external validity and reduced residual confounding by the active comparator.
⇒ Lack of additional data on some lifestyle risk factors.
⇒ Potential confounding introduced by ever users who received maintenance proton pump inhibitor use long before the study period.
⇒ Longer follow-up (ie, more than 10 years) may be needed.

many countries, with an estimated 10%–30% of adults regularly using PPIs.[3 4] Although PPIs have been deemed safe for short-term use and are recommended for a broad spectrum of indications by current guidelines and expert reviews,[5–7] only a few indications justify long-term use exceeding 4 weeks. Despite this, PPIs are often overprescribed,[7 8] which highlights the need for further investigation into the potential long-term side effects and public health implications of their widespread use.

Long-term PPI use has been associated with an increased risk of gastrointestinal cancers, including gastric, oesophageal and pancreatic cancers.[9–11] Previous studies have demonstrated that PPI use significantly impacts the gut microbiome and contributes to dysbiosis,[12–14] which has been linked to oncogenesis in the gut epithelium.[15 16] Enteric infections and the trophic effects of PPI-induced hypergastrinaemia may also serve as potential cancerogenic factors.[17] While some cohort studies have reported elevated relative risks of colorectal cancer (CRC) following PPI use,[18 19] others have found no association between CRC and PPI use.[2 20–25] However, potential methodological limitations in some of these studies have impacted the results. For example, some case–control studies introduced selection bias in exposure effect estimation by enrolling prevalent users rather

than new users,[21 22 24 26] and latency bias and time-window bias were present in some previous cohort studies due to the selection of narrow observation time ranges.[19 21 27]

To clarify the relationship between PPI use and CRC, we conducted a large cohort study using high-coverage national registries to control for immortal time bias and time-window bias. We aimed to assess the association between maintenance PPI use and the risk of colorectal adenocarcinoma (CRA), which accounts for >95% of all CRC cases.[28] We focused on right-sided and left-sided CRAs due to their chronological exposure to the drug-effects based on the anatomical locations.[28 29] Additionally, we used the less popular histamine-2 receptor antagonists (H₂RAs) as active comparators in the supplementary analysis to minimise confounding-by-indication, as they share similar indications with PPIs and were used for treating gastric acid-related disorders before the introduction of PPIs as more potent acid suppressants.[30]

## METHODS
### Study design
This was a register-based cohort study that enrolled all Swedish adults (aged 18 years old) who were on maintenance PPI therapy between July 2005 and December 2012. The study population was compared to the general population (ie, the Swedish total population) stratified by the same age, sex and calendar period. Participants were followed from 1 year after the first dispensed prescription date until the first diagnosis of CRA, death or the end of the study period (31 December 2012). Data were obtained from the Swedish Cancer Registry, the National Patient Registry, the Cause of Death Registry, the Prescribed Drug Registry and the Swedish Total Population Registry.[31–35] Cancer and demographic information of the total population can be downloaded from the National Board of Health and Welfare's statistical database.[31] The study protocol was established a priori, and the study was conducted in accordance with the Strengthening the Reporting of Observational Studies in Epidemiology (STROBE) statement.[36]

### Patient and public involvement
Due to its data-driven nature, this retrospective study had no patient or public directly involved in the study's design, data analysis, method selection or dissemination planning. Research questions and outcome measures were derived from existing literature and clinical relevance without direct input from them. Recruitment and follow-up relied on existing records from national registries.

### Exposure
PPI usage data were obtained from the Prescribed Drug Registry, labelled by the anatomical therapeutic chemical (ATC)/defined daily dose (DDD) system. This system annotated the DDD per prescribed package with ATC codes. PPIs are also available over the counter in Sweden

but only in small quantities and at exorbitant prices in comparison to prescribed PPIs.[37] Therefore, most PPI consumption of long-term users is likely recorded in this registry. Maintenance PPI use (ATC code A02BC) was defined as cumulative use for at least 180 days prior to the onset of any cancer, death, or the end of the study period. Maintenance H₂RA (ATC code A02BA) use served as an active comparator, and it was defined as using H₂RA for at least 180 days before the onset of any cancer, death, or the end of the study period.

### OUTCOME
The primary outcome was CRA, defined using the International Classification of Diseases (ICD, 10th edition) codes C18–C20 from the Swedish Cancer Registry,[38] and the histopathology code 096. The tumours were further classified based on their anatomical location as right-sided, left-sided or overlapping/unspecified sites (ICD codes in online supplemental table S1). The tumour-node-metastasis (TNM) classification of the American Joint Committee on Cancer eighth edition Staging of Colorectal Cancer was used to define tumour stage and grouped as follows: stages 0–I (Tis, N0, M0 for stage 0; T1–2, N0, M0 for stage I), II (T3–4, N0, M0), III (T1–4, N1–2, M0) and IV (any T, any N, M1).[39]

### Exclusions
Individuals with any documented cancer diagnosis prior to the study period or prior to the first dispensing date of PPIs were excluded. Individuals who switched between maintenance use of PPIs and H₂RAs (defined as ≥180 days of accumulated use of either drug) or had an exposure period of less than 1 year were also excluded. (online supplemental figure S1)

### Covariates
Age at treatment initiation was categorised into 18–39, 40–49, 50–59, 60–69, and ≥70 years old, and calendar periods were categorised into 2005–2006, 2007–2009, and 2010–2012. The duration of time since drug initiation was categorised as 1–2, 2–4, 4–6, and ≥6 years. No missing values for these variables were found for any of the study participants.

The most common recorded indications of maintenance PPI use were extracted from the National Patient Registry, including *Helicobacter pylori* infection or eradication, gastro-oesophageal reflux, Barrett's oesophagus, peptic ulcer, Zollinger Ellison's syndrome, gastro-duodenitis, dyspepsia, maintenance use (>180 days during the study period) of low-dose aspirin, and maintenance use of non-steroidal anti-inflammatory drugs (NSAIDs)[9 11 40] (ICD codes in online supplemental table S1). We performed subgroup analyses on individuals with and without the listed indications, as well as on each indication separately when the statistical power was sufficient.

### Statistical analyses
We calculated standardised incidence ratios (SIRs) with 95% confidence intervals (CIs) to assess the

association between maintenance PPI use and CRA. Standardisation was performed according to Breslow and Day's method, and the allocation of person-years followed Clayton's algorithm.[41] SIRs were obtained by calculating the ratio of observed CRA incidence rate compared to the expected incidence rate derived from the incidence among the Swedish total population of the same age, sex and calendar period. The information on the incidence of CRA (summarised in online supplemental table S2) and the demographic characteristics of the total population can be downloaded from the National Board of Health and Welfare's statistical database.[42] During the study period (2005–2012), the overall incidence of CRA in men ranged between 81.8 - 89.9/100 000 individuals and between 74.8 - 79.7/100 000 individuals in women (presented in online supplemental figure 2). In our main analysis, we compared the overall risk of CRA in the study population to that in the general population of Sweden, stratifying the main outcomes by tumour stages (0–I, II, III, IV, and missing values), tumour location (right-sided and left-sided), indications and follow-up time since drug initiation (1–2, 2–4, 4–6, and ≥6 years). We extensively investigated the risk of different tumour locations by age, sex, and indications for maintenance PPI use. Additionally, we assessed the risk of CRA in maintenance PPI users with a maximal follow-up time from the first dispensed prescription date, considering age, sex, calendar period, tumour stage and tumour location.

Furthermore, we conducted two supplementary analyses. First, we estimated SIRs for maintenance $H_2RA$ users and conducted a duration analysis (1–2, 2–4, 4–6, and ≥6 years). Second, we used a multivariable Poisson regression model to estimate incidence rate ratios (IRRs) with 95% CIs to compare CRA risk between maintenance PPI users and $H_2RA$ users, adjusting for age, sex, and common therapy indications.[43] The number of patients analysed for the Poisson regression model equals the summarised number of maintenance PPI users and maintenance $H_2RA$ users. Propensity score adjustment between maintenance PPI users and $H_2RA$ users was performed in the model to better control for confounders. Propensity scores were estimated using multivariable logistic regression, including age, sex, and indications. We applied $p$ values less than 0.05 statistically significant for all the analyses. Stata MP V.14.2 was used for all calculations.

## RESULTS
### Characteristics of the PPI cohort
This study included 754 118 adult maintenance PPI users with no prior history of cancer, with a median follow-up of 5.3 years (totalling 4 177 396 person-years) and a maximum follow-up of 7.5 years. Women (0.52%) had a lower cumulative incidence of CRA than men (0.62%). Maintenance PPI use was most common in

**Table 1** Characteristics of all maintenance proton pump inhibitor (PPI) users in Sweden between 2005 and 2012

| Characteristics of the study population | Maintenance PPI users | |
|---|---|---|
| | Number | % |
| Total | 754 118 | 100.0 |
| Age at starting the maintenance PPI use, years | | |
| 18–39 years | 84 510 | 11.2 |
| 40–49 years | 99 897 | 13.3 |
| 50–59 years | 149 167 | 19.8 |
| 60–69 years | 167 456 | 22.2 |
| ≥70 years | 253 088 | 33.6 |
| Sex | | |
| Men | 310 357 | 41.2 |
| Women | 443 761 | 58.9 |
| Calendar period | | |
| 2005–2006 | 430 527 | 57.1 |
| 2007–2009 | 221 369 | 29.4 |
| 2010–2012 | 102 222 | 13.6 |
| Colorectal adenocarcinoma | 4432 | 0.6 |
| Right-sided | 1884 | 0.3 |
| Left-sided | 2499 | 0.3 |
| Overlapping or unspecified | 49 | 0.0 |
| Tumour stages of colorectal adenocarcinoma | | |
| Stages 0–I | 674 | 0.1 |
| Stage II | 1102 | 0.1 |
| Stage III | 1017 | 0.1 |
| Stage IV | 748 | 0.1 |
| Missing | 891 | 0.1 |
| Indications for maintenance PPI use | | |
| *Helicobacter pylori* infection or eradication | 55 311 | 7.3 |
| Gastro-oesophageal reflux | 193 371 | 25.6 |
| Barrett's oesophagus | 5717 | 0.8 |
| Peptic ulcer | 74 946 | 10.0 |
| Zollinger Ellison's syndrome | 28 | 0.0 |
| Gastro-duodenitis | 100 191 | 13.2 |
| Dyspepsia | 41 916 | 5.6 |
| Aspirin maintenance use | 262 242 | 34.8 |
| Non-steroidal anti-inflammatory drugs maintenance use | 231 792 | 30.7 |
| Without any of these recorded indications | 186 529 | 24.7 |
| Maintenance PPI use based on daily drug usage | | |
| IQR | 1810 | – |
| Person-years | 3 426 925 | – |
| Follow-up time, years (median) | 5.3 | – |

IQR, interquartile range; PPI, proton pump inhibitor.

individuals aged 60–69 and ≥70 years old (tables 1 and 2). A total of 4432 cases of CRA were identified, with the majority being left-sided (n=2499, 56.4%), stages II (n=1102, 24.9%) and III (n=1017, 22.9%).

**Table 2** Risk of colorectal adenocarcinomas in maintenance proton pump inhibitor (PPI) users, expressed as standardised incidence ratios (SIRs) and 95% confidence intervals (CIs) by age, sex, calendar period, tumour subsites, and stages

| | Maintenance PPI users | |
|---|---|---|
| | Number | SIR (95% CI) |
| Total colorectal adenocarcinomas | 4432 | 1.10 (1.06 to 1.13) |
| Sex | | |
| Men | 2113 | 1.13 (1.08 to 1.18) |
| Women | 2319 | 1.10 (1.06 to 1.15) |
| Age at starting the maintenance PPI use, years | | |
| 18–39 | 38 | 2.79 (1.62 to 4.47) |
| 40–49 | 145 | 2.02 (1.65 to 2.45) |
| 50–59 | 531 | 1.27 (1.13 to 1.41) |
| 60–69 | 1277 | 1.26 (1.19 to 1.34) |
| ≥70 | 2441 | 1.04 (1.00 to 1.07) |
| Calendar year period | | |
| 2005–2006 | 3215 | 1.29 (1.09 to 1.51) |
| 2007–2009 | 1049 | 1.17 (1.12 to 1.22) |
| 2010–2012 | 168 | 1.07 (1.02 to 1.11) |
| Subsites of colorectal adenocarcinoma | | |
| Right-sided | 1884 | 1.26 (1.20 to 1.32) |
| Left-sided | 2499 | 0.99 (0.96 to 1.03) |
| Stages of colorectal adenocarcinoma | | |
| Stages 0–I | 674 | 1.22 (1.13 to 1.32) |
| Stage II | 1102 | 1.10 (1.03 to 1.16) |
| Stage III | 1017 | 1.07 (1.00 to 1.13) |
| Stage IV | 748 | 0.95 (0.88 to 1.02) |
| Missing | 891 | 1.19 (1.11 to 1.27) |

CI, confidence interval; PPI, proton pump inhibitor; SIR, standardised incidence ratio.

## Risk of colorectal adenocarcinoma

Compared to the general population, the overall risk of CRA was increased in maintenance PPI users (SIR 1.10, 95% CI 1.06 to 1.13). This risk was particularly affected by age, with the highest estimates present in those aged 18–39 years (SIR 2.79, 95% CI 1.62 to 4.47) and 40–49 years (SIR 2.02, 95% CI 1.65 to 2.45), while no convincing association was found in individuals ≥70 years old (SIR 1.04, 95% CI 1.00 to 1.07). In maintenance PPI users, the diagnosed CRA was particularly associated with an increased risk of early stage (stages 0–I, SIR 1.12, 95% CI 1.04 to 1.21) (table 2). In terms of tumour location, maintenance PPI users had a higher risk of right-sided CRA than the general

population, regardless of age and sex (tables 2 and 3). Meanwhile, no association between PPI use and CRA risk was found in left-sided CRC (tables 2 and 3).

## Common indications for maintenance PPI use

Indications for maintenance PPI use did not show the same trend for the drug–cancer–risk relationship in the exposed population. PPI use was not related to an increased risk of CRA among patients with Barrett's oesophagus. A marginally increased risk of CRA among PPI users was observed in participants with the indications of *Helicobacter pylori* infection or eradication (SIR 1.29, 95% CI 1.15 to 1.44), dyspepsia (SIR 1.29, 95% CI 1.16 to 1.42), gastro-duodenitis (SIR 1.28, 95% CI 1.19 to 1.38), gastro-oesophageal reflux (SIR 1.26, 95% CI 1.19 to 1.33) and peptic ulcer (SIR 1.16, 95% CI 1.07 to 1.26). Among individuals with the combined use of non-steroidal anti-inflammatory drugs (NSAIDs) or low-dose aspirin, maintenance PPI use was not significantly associated with the risk of CRA (table 4). Compared to the general population, most of the indications were related to an increased risk of right-sided CRAs in maintenance PPI users (table 4).

## Duration of follow-up time

There was an elevated risk of CRA observed between the exposed population and the general population during the 1–2 years of follow-up (SIR 1.24, 95% CI 1.17 to 1.32). Thereafter, the results suggested slightly decreased risks of CRA in the 2–4 years (SIR 0.91, 95% CI 0.87 to 0.96), 4–6 years (SIR 0.65, 95% CI 0.61 to 0.68), and ≥6 years (SIR 0.85, 95% CI 0.77 to 0.92) of follow-up (figure 1A). There was no obvious difference in CRA risks observed between the maintenance $H_2$RA users and the general population during the follow-up time (figure 1A). An increased risk of right-sided CRA was also observed during the 1–2 years of follow-up (figure 1B).

## Comparison with maintenance $H_2$RA use

With a total of 19 795 individuals and 130 810 person-years of follow-up, the risk of CRA was not found to be associated with maintenance $H_2$RA use compared to the background population (n=115, SIR 0.85, 95% CI 0.71 to 1.03). The multivariable Poisson regression model of maintenance PPI versus $H_2$RA use showed an IRR of 1.05 (95% CI 0.87 to 1.27, $p<0.05$).

## DISCUSSION

In this large population-based study investigating the association between maintenance PPI use and CRC risk, we found a potential 10% higher risk of CRA compared to the general population. SIRs showed that maintenance PPI use might be associated with an increased risk of right-sided CRA compared to that in the general population. Young to middle-aged adult (ie, 18–39 and 40–49 years old) participants showed a higher risk of CRA than the general population.

**Table 3** Sex and age disparities in the risk of right-sided and left-sided colorectal adenocarcinomas in maintenance proton pump inhibitor (PPI) users, expressed as standardised incidence ratios (SIRs) and 95% confidence intervals (CIs)

| Cancer in categories | Right-sided colorectal adenocarcinoma | | Left-sided colorectal adenocarcinoma | |
|---|---|---|---|---|
| | Number | SIR (95% CI) | Number | SIR (95% CI) |
| Sex | | | | |
| Men | 736 | 1.25 (1.16 to 1.34) | 1359 | 1.04 (0.98 to 1.10) |
| Women | 1148 | 1.27 (1.19 to 1.34) | 1,14 | 0.94 (0.89 to 1.00) |
| Age at starting the maintenance PPI, years | | | | |
| 18–39 | 13 | 2.78 (1.02 to 6.06) | 24 | 2.81 (1.40 to 5.03) |
| 40–49 | 37 | 1.98 (1.31 to 2.89) | 103 | 1.95 (1.53 to 2.46) |
| 50–59 | 167 | 1.42 (1.14 to 1.75) | 359 | 1.20 (1.05 to 1.37) |
| 60–69 | 528 | 1.58 (1.43 to 1.75) | 732 | 1.12 (1.03 to 1.21) |
| ≥70 | 1139 | 1.17 (1.11 to 1.24) | 1281 | 0.90 (0.86 to 0.95) |

CI, confidence interval; PPI, proton pump inhibitor; SIRs, standardised incidence ratios.

Our findings are in line with an estimated exposure effect of 2.03 (95% CI 1.56 to 2.63) between PPI users and CRC risk reported by a Taiwanese cohort,[18] where immortal time bias could not be ruled out. In our study, we excluded 41 697 individuals with less than 1 year of follow-up to avoid detection bias. As CRA typically takes several to 20 years to develop from polyp to adenocarcinoma,[44] cancer diagnoses or deaths within 1 year after new incident PPI prescriptions should be excluded because they could result in a biological effect indicating that PPI initiation preceded or increased the cancer risks. We have further carefully addressed the possibility that our strict exclusion of the first year of follow-up introduced immortal time bias. Therefore, we estimated the overall risk of CRA within the same study population starting follow-up from the first dispensed prescription date of PPIs. The results were consistent with our main analysis (online supplemental table S3), suggesting that immortal time bias might not be a problem; otherwise, the overestimates of healthy individuals should have led to an opposite observed effect.

In the duration analysis, maintenance PPI use was associated with an increased risk of CRA than the general population during the 1-2 years of follow-up. Following that the exposed population showed a slightly lower risk of CRA than the general population. Then, the exposed population had a lower but less decreased risk of CRA compared to the general population with ≥6 years of follow-up. A similar pattern was observed for the risk of right-sided CRA. This may be because of potential confounding introduced by ever users who received maintenance PPI use long before the study period, as we do not have information on prescribed drug use prior to the start of the study (July 2005) when the National Prescribed Drug Registry was not fully constructed. A longer follow-up (ie, more than 10 years) and more extensive wash-out of ever users may be needed, so that an increasing risk of CRA would be observed. Meanwhile, reverse causality (ie, the observed effect explained by reversely linking initiating PPI to subclinical symptoms of CRC) should also be considered during the follow-up. If CRC patients were prone to be prescribed maintenance-used PPIs, substantial positive effects on the exposure estimation might have influenced each stratum of the exposed population during our analysis. However, the different trends of right-sided versus left-sided CRC risk during different time periods (figure 1B) and with different indications (table 4) indicated that the reverse causality due to the prevalent use of maintenance PPIs might be less concerning.

Among the indication subgroup analyses, only combined maintenance PPI use with NSAIDs or low-dose aspirin showed a potential preventive effect against colorectal tumour progression. A previous American cohort study concluded that long-term use of NSAIDs and low-dose aspirin reduced the risk of CRC.[45] Our study also provided evidence that long-term NSAID or low-dose aspirin use appeared to be beneficial in concomitant use with PPIs. While we acknowledged the existing literature of concomitant drug use with PPIs, the necessity for continued evaluation in light of the increased focus on combined PPI use with NSAIDs or low-dose aspirin still needs to be emphasised in assessing CRC preventive potential. Nevertheless, an increased risk of CRA was observed among the *Helicobacter pylori* infection or eradication, gastro-oesophageal reflux, gastro-duodenitis and dyspepsia groups, which have not been established as the risk factors for CRA. The elevated risk could be explained by the fact that acid-related disorders facilitate changes in the upper-intestinal microbiome barrier owing to gastric acid over secretion.[46 47] Gut dysbiosis and hypergastrinaemia may also contribute to the association between PPI use and colorectal carcinogenesis.[15 48 49]

The strengths of our study include not only the high number of individuals and person-years, but also the well-designed method for calculating cancer risks. The high-quality, high-coverage national registries facilitated the identification of eligible individuals with a lengthy follow-up. Compared with previous studies, we constructed this cohort with many individuals and long follow-up times to minimise time-related bias. Moreover, the results are generalisable to other Nordic nations

**Table 4** Risk of (**a**) colorectal adenocarcinomas (**b**) right-sided and left-sided colorectal adenocarcinomas according to indications for maintenance proton pump inhibitor (PPI) users, expressed as standardised incidence ratios (SIRs) and 95% confidence intervals (CIs)

**(a) Risk of colorectal adenocarcinomas**

| Indications for maintenance of PPI use | Colorectal adenocarcinoma | |
|---|---|---|
| | Number of cases (% in maintenance PPI users) | SIR (95% CI) |
| Without any below indication | 879 (4.7) | 1.13 (1.06 to 1.21) |
| *Helicobacter pylori* infection or eradication | 328 (5.9) | 1.29 (1.15 to 1.44) |
| Gastro-oesophageal reflux | 1352 (7) | 1.26 (1.19 to 1.33) |
| Barrett's oesophagus | 39 (6.8) | 1.16 (0.83 to 1.59) |
| Peptic ulcer | 625 (8.3) | 1.16 (1.07 to 1.26) |
| Zollinger Ellison's syndrome | 0 | – |
| Gastro-duodenitis | 739 (7.4) | 1.28 (1.19 to 1.38) |
| Dyspepsia | 368 (8.8) | 1.29 (1.16 to 1.42) |
| Maintenance use of low-dose aspirin | 2026 (7.7) | 1.00 (0.96 to 1.05) |
| Maintenance use of non-steroidal anti-inflammatory drugs | 1096 (4.7) | 1.00 (0.94 to 1.06) |
| With any of the above indications | 3553 (6.3) | 1.09 (1.05 to 1.12) |

**(b) Risk of right-sided and left-sided colorectal adenocarcinomas**

| Indications for maintenance of PPI use | Right-sided colorectal adenocarcinoma | | Left-sided colorectal adenocarcinoma | |
|---|---|---|---|---|
| | Number of cases (% in maintenance PPI users) | SIR (95% CI) | Number of cases (% in maintenance PPI users) | SIR (95% CI) |
| Without any below indication | 360 (1.9) | 1.27 (1.15 to 1.41) | 511 (2.7) | 1.05 (0.97 to 1.15) |
| *Helicobacter pylori* infection or eradication | 148 (2.7) | 1.63 (1.37 to 1.91) | 176 (3.2) | 1.09 (0.94 to 1.27) |
| Gastro-oesophageal reflux | 616 (3.2) | 1.58 (1.45 to 1.70) | 718 (3.7) | 1.06 (0.99 to 1.15) |
| Barrett's oesophagus | 17 (3) | 1.53 (0.89 to 2.45) | 21 (3.7) | 0.95 (0.59 to 1.45) |
| Peptic ulcer | 288 (3.8) | 1.45 (1.28 to 1.62) | 333 (4.4) | 1.00 (0.90 to 1.11) |
| Zollinger Ellison's syndrome | 0 | – | 0 | |
| Gastro-duodenitis | 361 (3.6) | 1.70 (1.53 to 1.88) | 370 (3.7) | 1.03 (0.93 to 1.14) |
| Dyspepsia | 186 (4.4) | 1.74 (1.50 to 2.01) | 179 (4.3) | 1.01 (0.87 to 1.17) |
| Maintenance use of low-dose aspirin | 900 (3.4) | 1.18 (1.10 to 1.26) | 1110 (4.2) | 0.90 (0.85 to 0.95) |
| Maintenance use of non-steroidal anti-inflammatory drugs | 397 (1.7) | 0.98 (0.89 to 1.09) | 685 (3) | 1.00 (0.92 to 1.08) |
| With any of the above indications | 1524 (2.7) | 1.26 (1.19 to 1.32) | 1988 (3.5) | 0.98 (0.94 to 1.02) |

CI, confidence interval; PPI, proton pump inhibitor; SIRs, standardised incidence ratios.

because comparable recording systems are maintained in other Nordic nations. The method enables us to compare large populations at different levels without considering migrations and drop-offs. In addition, we accounted for an alternative indicator of $H_2RA$ to reduce the residual confounding caused by PPI indications, which could lead to bidirectional results in the association between exposure and outcome. $H_2RAs$ are also used as acid suppressants to treat stomach acid production conditions like peptic ulcers and Zollinger-Ellison syndrome, and provide relief from excessive acid secretion symptoms.[50] Similar to PPI, prolonged use of $H_2RAs$ can lead to elevated systemic levels of gastrin, a hormone known to stimulate the proliferation of colorectal epithelium and contribute to colon adenoma progression.[50 51] Yet, our study showed that $H_2RAs$ may not be associated with CRA/CRC risks, in accordance with previous population-based studies.[21 52] Thus, the residual

confounding by indication was minimised when comparing maintenance PPI and maintenance $H_2RA$ use.

Our study has certain limitations, including a lack of additional data on lifestyle risk factors such as smoking, alcohol, diet, and body mass index. Moreover, although we followed the participants up to a maximum duration of 7.5 years, a median follow-up time of 5.3 years may be insufficient. A longer median follow-up permits a more extended observation of all naturally evoked outcomes and a more extensive exclusion of prevalent use without harming the sample size and power too much. Nevertheless, during the long progression period, the clinical diagnosis of CRA can be delayed following the onset of cancerous symptoms, which will dilute the association between the PPI and CRC risk.

A

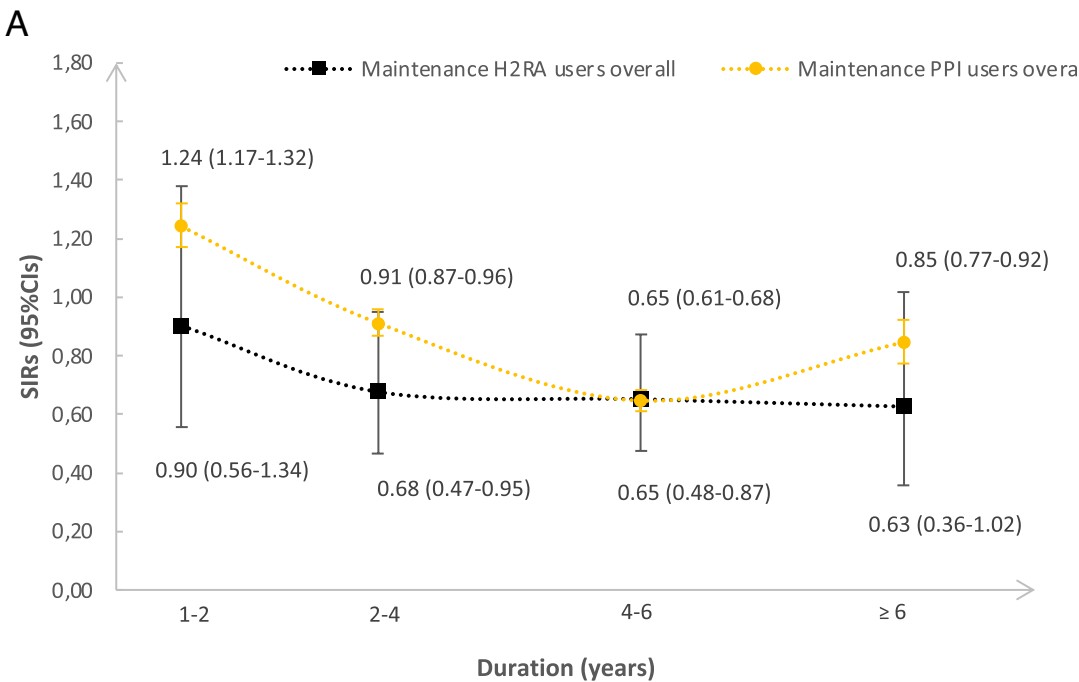

B

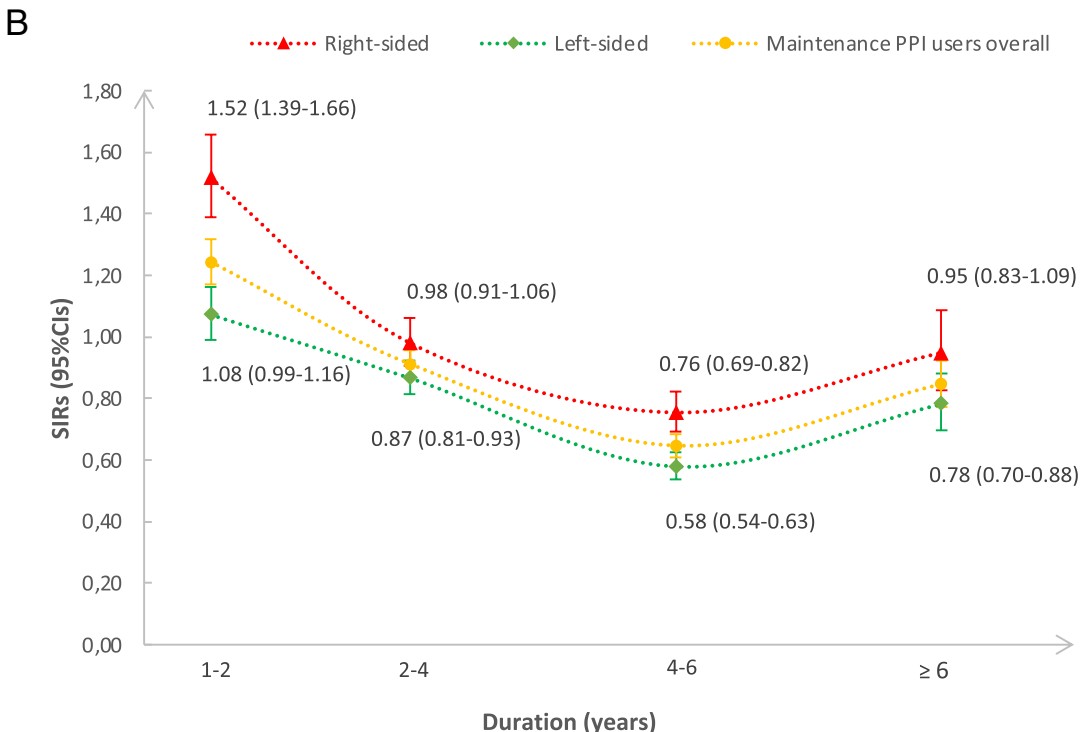

**Figure 1** Duration of follow-up and risk of colorectal adenocarcinomas among (A)maintenance proton pump inhibitor (PPI) users and maintenance histamine-2 receptor antagonist (H₂RA) users (B)overall maintenance PPI users and users diagnosed with right-sided or left-sided colorectal adenocarcinomas, expressed as standardised incidence ratios (SIRs) and 95% confidence intervals (CIs).

In our study, maintenance PPI users account for less than 11% of the Swedish population (7.1–7.6 million people).[53 54] The prevalence is lower than that of other Nordic countries, such as 15.5% in Denmark.[3] But inappropriate PPI prescriptions are increasing among adults.[55 56] Meanwhile, early onset CRC (diagnosed at <50 years of age) has been reported to be rising, especially as young populations at increased risk of CRC tend to have an unhealthy diet, a lack of exercise, obesity and a low acceptance of colonoscopy.[57–59]

The message to clinics is that young to middle-aged people should pay attention to the potential risk of CRA associated with maintenance PPI use. Moreover, combining low-dose aspirin or other NSAIDs with maintenance PPI treatment could be meaningful to PPI users with common indications, although further studies are needed before any firm conclusions can be made. Finally, right-sided CRA is more clinically insidious, more advanced at diagnosis and worse in prognosis than left-sided CRA.[29 60 61] Early chemoprevention for maintenance PPI users should be considered to prevent poorer outcomes of right-sided CRA because of the significantly increased risk of right-sided CRA following maintenance PPI use observed in our study than the general population.

In conclusion, our study showed that maintenance PPI use may be associated with an increased risk of CRA. Nevertheless, we need more population-based studies with extended observation to confirm the association.

**Author affiliations**
[1]Department of Microbiology, Tumour and Cell Biology, Karolinska Institutet, Stockholm, Sweden
[2]Centre for Psychiatry Research, Department of Clinical Neuroscience, Karolinska Institutet, Stockholm, Sweden
[3]Department of Clinical Sciences, Intervention and Technology, Karolinska Institutet, Stockholm, Sweden
[4]Department of Surgery, Capio Saint Göran Hospital, Stockholm, Sweden
[5]Clinical Epidemiology and Biostatistics School of Medical Sciences, Örebro University, Örebro, Sweden
[6]Institute of Environmental Medicine, Karolinska Institutet, Stockholm, Sweden
[7]Department of Head and Skin, Ghent University, Ghent, Belgium
[8]Global Health Institute, Antwerp University, Antwerp, Belgium

**Acknowledgements** We wish to address our gratitude to the Swedish National Board of Health and Welfare (Socialstyrelsen) for collecting all the data and to all the participants who contributed to the data collection.

**Contributors** All authors designed the study and approved the final version of the manuscript (QL, XW, LE, OS, KF and NB). QL drafted the protocol, conducted the statistical analysis and wrote the draft of the manuscript, which was revised and approved by all the other authors. NB was the guarantor of the study. QL and NB had full access to all the data and codes in the study.

**Funding** This study was sponsored by Svenska Läkaresallskapet (SLS-788731, SLS-788751 and SLS-783091) and the Swedish Research Council (2020-01058). QL was supported by a China Scholarship Council Grant (201700260302). None of these sponsors had any role in the study design, the collection, the analysis or the interpretation of data.

**Competing interests** None declared.

**Patient and public involvement** Patients and/or the public were not involved in the design, or conduct, or reporting or dissemination plans of this research.

**Patient consent for publication** Not applicable.

**Ethics approval** This study involves human participants and ethical approval was obtained from the Regional Ethical Review Board in Stockholm (2014/1291-31/4). No individual informed consent was required due to the registry-based nature of the data.

**Provenance and peer review** Not commissioned; externally peer reviewed.

**Data availability statement** Data are available upon reasonable request. The data sets analysed during the current study are not publicly available due to the sharing agreement at Karolinska Institutet and the National Board of Health and Welfare but are available from the corresponding authors on reasonable request.

**ORCID iDs**
Qing Liu http://orcid.org/0000-0002-1663-189X
Nele Brusselaers http://orcid.org/0000-0003-0137-447X

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
