## [Reviewer comments · BMJ Open]

ARTICLE DETAILS

TITLE (PROVISIONAL)	Maintenance proton pump inhibitor use and risk of colorectal cancer: A Swedish retrospective cohort study
AUTHORS	Liu, Qing; WANG, Xinchun; Engstrand, Lars; Sadr-Azodi, Omid; Fall, Katja; Brusselaers, Nele

VERSION 1 – REVIEW

REVIEWER	Waldum, Helge Norwegian University of Science and Technology, Department of Cancer Research and Molecular Medicine
REVIEW RETURNED	27-Nov-2023

GENERAL COMMENTS	Based on Swedish registries the possible role of proton pump inhibitor (PPI) treatment in colonic carcinogenesis is examined. Long-term PPI treatment was defined as more than 180 days and the occurrence of adenocarcinoma of the colon registered for a maximal time of 7 years. The first year after the PPI treatment was excluded, and the PPI exposed individuals were evaluated in the total material as well as according to sex, age at PPI use, indication, location of the cancer and observation time compared with the general population and a group of subjects using histamine-2 receptor antagonist (H2RA). They found a slight increased risk of colonic cancer in those having used PPI in the total material, in both sexes and in most indications. It is, however, peculiar that younger persons had a higher risk of colon cancer after PPI use than older. Could that be due to less suspicion of cancer in the young groups which accordingly were given PPI without adequate examination? The higher risk for colonic cancer in right-sided compared with left-sided can also be due longer time from symptom debut and diagnoses (Reverse causality)? It is interesting that Aspirin and Non-steroidal anti-inflammatory drugs makes up 65 % of the total material. This should be commented on. Although there exist previous studies showing that PPI use in those patients using such drugs, with the increased focus on the side effects of PPIs, this should probably be reevaluated. Finally, it is not likely that dysbiosis should be the cause of the possible colonic carcinogenic effect of PPIs. However, it is possible that taking away the killing of microbes (not only bacteria) due to loss of gastric juice function, could allow a specific carcinogen escape to the gut.
--

REVIEWER	Dai, Haibin Zhejiang University School of Medicine Second Affiliated Hospital, Pharmacy
REVIEW RETURNED	21-Dec-2023

GENERAL COMMENTS	1. Page 6 Line 3-5, The study population was compared to the entire Swedish population stratified by the same age, sex, and calendar period: the authors used the prevalence of CRA in the general population as a baseline for comparison with patients on long-term PPI use. In this manuscript, the general population meant people who received at least one commonly used drug prescription [FigureS1(a)]. Are people prescribed at least one common medication equal to the total population? 2. Page 8 Line 12-13, Page 27 Figure S1(a): The authors selected a total sample size of 843,003, of which 754,118 patients were included in the study, meaning that 89.5% were maintenance PPI users. Is the percentage of maintenance PPI users in this sample too high? 3. Do two factors in question 1 and 2 lead to bias in research results? 4. Why did the authors not consider PPI non-users or PPI irregular users (≤ 180 days) as a control group? 5. There is no information on the total population (incidence of CRA, demographic characteristics, etc.). 6. Page 3 Line 17-19: The authors defined maintenance PPI users as at least 180 days of usage. However, the maximum follow-up period was 7.5 years, and patients could theoretically use PPIs daily to once every 17.2 days. Is there any basis for this definition of maintenance PPI users? 7. Page 6 Line 41-42: Although Figure S1 (b) mentioned needing medication for >180 days, there was no definition of maintenance H2RA in the manuscript. 8. There is a significant difference in the number of maintenance users of PPI and H2RA in this manuscript. Is the therapeutic effect of PPI and H2RA equal in the clinic? Are there differences in disease type and severity in the treated population between these two classes of medications? 9. The PPI maintenance users may have gastrointestinal symptoms or diseases, compared with general populations, and H2RA users how these gastrointestinal symptoms or diseases are related to CRA is recommended to discuss the diagnosis of PPI medication in detail on a case-by-case basis.
---

REVIEWER	Knapp, Guido Technische Universität Dortmund
REVIEW RETURNED	02-Feb-2024

GENERAL COMMENTS	The authors used a retrospective cohort study to investigate the risk of colorectal cancer associated with long-term use of proton pump inhibitors. The section on statistical analyses clearly describes the methods used. Maybe the authors should add a sentence or two about the propensity score adjustment. No information about the realization is given here. Table 3: The lower forest plot is too high (between the lines). And most likely shifted to the right hand side, for instance, last confidence interval does not include one. Please correct! Results 3. Common indications for maintenance PPI use: Confidence interval for dyspepsia in the text is not correct. Table 4: Please check again the position of the forest plots!
--

	It is unclear how many patients have been analyzed for the Poisson regression model. Please add the information!
--	--

VERSION 1 – AUTHOR RESPONSE

Reviewer: 1

Dr. Helge Waldum, Norwegian University of Science and Technology

Comments to the Author:

Based on Swedish registries the possible role of proton pump inhibitor (PPI) treatment in colonic carcinogenesis is examined. Long-term PPI treatment was defined as more than 180 days and the occurrence of adenocarcinoma of the colon registered for a maximal time of 7 years. The first year after the PPI treatment was excluded, and the PPI exposed individuals were evaluated in the total material as well as according to sex, age at PPI use, indication, location of the cancer and observation time compared with the general population and a group of subjects using histamine-2 receptor antagonist (H2RA). They found a slight increased risk of colonic cancer in those having used PPI in the total material, in both sexes and in most indications. It is, however, peculiar that younger persons had a higher risk of colon cancer after PPI use than older. Could that be due to less suspicion of cancer in the young groups which accordingly were given PPI without adequate examination? The higher risk for colonic cancer in right-sided compared with left-sided can also be due longer time from symptom debut and diagnoses (Reverse causality)? It is interesting that Aspirin and Non-steroidal anti-inflammatory drugs makes up 65 % of the total material. This should be commented on. Although there exist previous studies showing that PPI use in those patients using such drugs, with the increased focus on the side effects of PPIs, this should probably be reevaluated. Finally, it is not likely that dysbiosis should be the cause of the possible colonic carcinogenic effect of PPIs. However, it is possible that taking away the killing of microbes (not only bacteria) due to loss of gastric juice function, could allow a specific carcinogen escape to the gut.

>>Reply:

Thank you for your insightful and constructive comments on our manuscript. We appreciate the thoughtful considerations and have addressed your points with the following responses:

- Age-related relative risk of colon cancer after PPI use: We agree with your point regarding the age-related cancer risk. Reverse causality may be more common in younger individuals, as physicians may prescribe PPIs with fewer clinical examinations for vague gastrointestinal symptoms to a young individual (1), meaning PPIs are prescribed for yet undiagnosed early cancer signs. We expect younger PPI users to have fewer comorbidities than their older counterparts (lower absolute risks), yet they will have more comorbidities than peers of the same age (higher SIRs). In our manuscript, the risk of colorectal adenocarcinoma (CRA) decreased relatively by age, with the highest SIRs in those younger than 40 years old- while the absolute risk of CRA among young individuals is low (N= 38 in <40 years). This trend was also seen in our previous association studies evaluating PPIs and the risk of gastric, oesophageal, biliary, and pancreatic cancer (2-5). Suppose this finding reflects a true causal association, PPIs should create a more harmful (cancer-promoting) environment in younger individuals than in older ones. Yet, this finding could also imply residual confounding (e.g., by smoking, lifestyle, diet, or others). There may also be some birth-cohort effect as the oldest individuals could not have been exposed to PPIs at the same young age (PPIs were first commercialised in the 1980s) but have had other lifestyle/diet exposures earlier in life.
- Right-sided vs. Left-sided colonic cancer risk: We acknowledged your concerns regarding the different timing of right-sided and left-sided colon cancer diagnoses. Indeed, a longer diagnostic time window may cause patients to be exposed to more or earlier PPI prescriptions. Right-sided CRA is more clinically insidious/asymptomatic than left-sided CRA (6), leading to a shorter diagnostic time window and a lower diagnostic rate of right-sided CRA if not interfered with. In this study, we

observed a higher risk of right-sided colonic cancer than left-sided. If reverse causality influenced the outcomes, this finding would indicate the opposite. Thus, this finding suggests that reverse causality due to symptom debut and diagnosis of sublocations may not be a concern in this study.

- Concomitant use of aspirin or NSAIDs with PPIs: Thank you for pointing this out. We have now defined low-dose aspirin and NSAID maintenance use as “> 180 days during the study period” in the Covariates section. We need to clarify that the concomitant use of aspirin with PPIs and concomitant use of NSAIDs with PPIs were not mutually exclusive when data was extracted from the National Prescribed Drug Registry using the ATC/DDD system. Thus, we may not add them up to 65% directly. Moreover, the percentages of concomitant PPI and aspirin or NSAID use have been published elsewhere (2,3), indicating the circumstances of PPI use in Swedish adults during our study period. Also, our findings were consistent with a previously published Swedish cohort concluding that both aspirin and NSAID users who used PPIs had a greater risk of developing all forms of gastrointestinal cancer (7). Still, in the revised discussion, we provided a detailed commentary on this aspect: “While we acknowledged the existing literature of the concomitant drug use of PPIs, the necessity for continued evaluation in light of the increased focus on combined PPI use with NSAIDs or low-dose aspirin still needs to be emphasised in assessing CRC preventive potential.”

- Dysbiosis and colonic carcinogenic effect of PPIs: Thanks for your insight into the connection between dysbiosis and the potential “cancerous effect” of PPIs. Although our study may not provide direct evidence on this causative chain (such as pH change of the GI content), it is essential to consider the different underlying mechanisms elucidating the complex interplay between PPI use, dysbiosis, and colonic carcinogenesis. There’s also evidence indicating that dysbiosis occurs when the gut loses protective bacteria and becomes overloaded with pathogenic and cancer-promoting bacteria, boosting cancer-related processes such as angiogenesis, apoptosis loss, and cell proliferation (8). Therefore, we cautiously conclude that maintenance PPI use may be associated with a higher colorectal adenocarcinoma risk but not with a causative cancerous effect.

We believe these revisions enhance the robustness and clarity of our manuscript. We value your feedback and welcome any additional suggestions you may have. Thank you again for your time and expertise.

References:

- (1) Deeks, A., Lombard, C., Michelmore, J. et al. The effects of gender and age on health related behaviors. *BMC Public Health* 9, 213 (2009).
- (2) Brusselaers, N., Sadr-Azodi, O., & Engstrand, L. (2020). Long-term proton pump inhibitor usage and the association with pancreatic cancer in Sweden. *Journal of Gastroenterology*, 55(4), 453–461.
- (3) Kamal, H., Sadr-Azodi, O., Engstrand, L., & Brusselaers, N. (2021). Association Between Proton Pump Inhibitor Use and Biliary Tract Cancer Risk: A Swedish Population-Based Cohort Study. *Hepatology*, 0(0), 2021.
- (4) Brusselaers, N., Engstrand, L., & Lagergren, J. (2018). Maintenance proton pump inhibition therapy and risk of oesophageal cancer. *Cancer Epidemiology*, 53(November 2017), 172–177.
- (5) Brusselaers, N., Wahlin, K., Engstrand, L., & Lagergren, J. (2017). Maintenance therapy with proton pump inhibitors and risk of gastric cancer: A nationwide population-based cohort study in Sweden. *BMJ Open*, 7(10), e017739.
- (6) Baran B, Mert Ozupek N, Yerli Tetik N, Acar E, Bekcioglu O, Baskin Y. Difference Between Left-Sided and Right-Sided Colorectal Cancer: A Focused Review of Literature. *Gastroenterology Res.* 2018 Aug;11(4):264-273.
- (7) Brusselaers N, Lagergren J. Maintenance use of non-steroidal anti-inflammatory drugs and risk of gastrointestinal cancer in a nationwide population-based cohort study in Sweden. *BMJ Open.* 2018 Jul 7;8(7):e021869.
- (8) Artemev A, Naik S, Pougno A, Honnavar P, Shanbhag NM. The Association of Microbiome Dysbiosis With Colorectal Cancer. *Cureus.* 2022 Feb 12;14(2):e22156.

Reviewer: 2

Dr. Haibin Dai, Zhejiang University School of Medicine Second Affiliated Hospital

Comments to the Author:

1. Page 6 Line 3-5, The study population was compared to the entire Swedish population stratified by the same age, sex, and calendar period: the authors used the prevalence of CRA in the general population as a baseline for comparison with patients on long-term PPI use. In this manuscript, the general population meant people who received at least one commonly used drug prescription [FigureS1(a)]. Are people prescribed at least one common medication equal to the total population?

>>Reply:

We apologise for this confusion. This referred to the design of the whole project, which also includes the evaluation of other drugs (menopausal hormones, NSAIDs, etc.). We have removed this information to clarify the design of this cohort, in which our exposed group received at least 6 months of PPIs (accumulated exposure). The comparison group was the entire Swedish population, which was used to compare the risk for the same age, sex, and calendar period.

2. Page 8 Line 12-13, Page 27 Figure S1(a),: The authors selected a total sample size of 843,003, of which 754,118 patients were included in the study, meaning that 89.5% were maintenance PPI users. Is the percentage of maintenance PPI users in this sample too high?

>>Reply:

To study a population with prolonged PPI use due to its clinical significance, we included all Swedish people who had maintenance PPIs during the study period regardless of PPI-only use or combined use. Meanwhile, we have applied strict inclusion and exclusion criteria to justify the sample composition. Thus, the percentage of maintenance PPI users is valid and accurate. We have added the prevalence of use in the discussion as: "In our study, maintenance PPI users account for less than 11% of the Swedish population (7.1-7.6 million people)" (9,10). "The prevalence is lower than that of other Nordic countries, such as 15.5% in Denmark" (11).

3. Do two factors in question 1 and 2 lead to bias in research results?

>>Reply:

Thank you for your comments on our study population. We have carefully considered your points and made clarifications. SIRs calculation and the background population helped to minimize selection bias, so they would not cause any more concern in our study results. However, we may have other concerns such as information bias due to the over-the-counter (OTC) PPIs in Sweden. And we cannot fully address lifestyle-related confounding from the national registries because of the retrospective data extraction process. We included these biases in our discussion.

4. Why did the authors not consider PPI non-users or PPI irregular users (≤ 180 days) as a control group?

>>Reply:

We appreciate your suggestion regarding the inclusion of PPI non-users or irregular users as a control group. We opted for a relatively high cut-off of exposure (180 days accumulated use), as other studies used a minimal PPI use of 1-2 prescriptions, to ensure a relatively homogenous group reflecting maintenance/chronic use. In addition, we are bound to the data availability restrictions, as we only have health registry information on approximately 85% of the total Swedish population (based on the use of selected commonly prescribed drugs) (10). We can't distinguish between this background population that was exposed and those that were unexposed. Also, information on PPI non-users or irregular users cannot be entirely recorded in the national registries due to the

accessibility of OTC PPIs in Sweden. On the contrary, maintenance PPI users were most likely to be recorded due to the reimbursement policy of prescribed drugs (stated in the Exposure section). Thus, our research question focused mainly on PPI users and the general population.

5. There is no information on the total population (incidence of CRA, demographic characteristics, etc.).

>>Reply:

We acknowledged your point about the need for information on the total population. The information on the incidence of CRA and the demographic characteristics of the total population can be downloaded from the National Board of Health and Welfare's statistical database (12), an open resource. We have stated it in the Statistical analyses section and added a supplementary table S3 in the revised manuscript and supplementary materials.

6. Page 3 Line 17-19: The authors defined maintenance PPI users as at least 180 days of usage. However, the maximum follow-up period was 7.5 years, and patients could theoretically use PPIs daily to once every 17.2 days. Is there any basis for this definition of maintenance PPI users?

>>Reply:

As mentioned above, we intended to investigate the long-term effect of PPI usage, especially beyond the guideline use (> 8 weeks). In previously published studies, we also estimated the accumulated \geq 180 days of drug usage based on the ATC/DDD system, which is an average estimate of drug prescriptions (2-5).

7. Page 6 Line 41-42: Although Figure S1 (b) mentioned needing medication for >180 days, there was no definition of maintenance H2RA in the manuscript.

>>Reply:

We have now included a clear definition of maintenance H2RA use in the Exposure section: "Maintenance H2RA (ATC code A02BA) use served as an active comparator, and it was defined as using H2RA for at least 180 days before the onset of any cancer, death, or the end of the study period." Thank you for pointing this out.

8. There is a significant difference in the number of maintenance users of PPI and H2RA in this manuscript. Is the therapeutic effect of PPI and H2RA equal in the clinic? Are there differences in disease type and severity in the treated population between these two classes of medications?

>>Reply:

In Sweden, H2RAs are rarely prescribed, while PPIs are dominantly prescribed. The significant difference in the number of maintenance users of PPI and H2RA exists because of the clinical application of the drug. PPIs were considered more potent than H2RAs, but they shared the gastrointestinal indications in clinics, making H2RA the best available comparators in our study design because we can't obtain the information of non-users with gastrointestinal symptoms from the background population. We stated it in the Introduction section as "Additionally, we used the less popular histamine-2 receptor antagonists (H2RAs) as active comparators in the supplementary analysis to minimise confounding-by-indication, as they share similar indications with PPIs and were used for treating gastric acid-related disorders before the introduction of PPIs as more potent acid suppressants".

9. The PPI maintenance users may have gastrointestinal symptoms or diseases, compared with general populations, and H2RA users how these gastrointestinal symptoms or diseases are related to CRA is recommended to discuss the diagnosis of PPI medication in detail on a case-by-case basis.

>>Reply:

The listed gastrointestinal symptoms or diseases in indications shared by PPI and H2RA users were selected based on previous literature. We may not compare CRA risk in each stratum by indications of maintenance PPI use and H2RA use because it's a bit beyond our study question (additional models may be required). But in the revised discussion, we have expanded the discussion to address the relationship between H2RA use and colorectal adenocarcinomas: "H2RAs are also used as acid suppressants to treat stomach acid production conditions like peptic ulcers, Zollinger-Ellison syndrome, and provide relief from excessive acid secretion symptoms. Similar to PPI, prolonged use of H2RAs can lead to elevated systemic levels of gastrin, a hormone known to stimulate the proliferation of colorectal epithelium and contribute to colon adenoma progression. Yet, our study showed that H2RAs may not be associated with CRA/CRC risks, in accordance with previous population-based studies."

References:

(9) Ludvigsson JF, Almqvist C, Bonamy AK, Ljung R, Michaëlsson K, Neovius M, Stephansson O, Ye W. Registers of the Swedish total population and their use in medical research. *Eur J Epidemiol.* 2016 Feb;31(2):125-36.

(10) <https://www.scb.se/en/finding-statistics/statistics-by-subject-area/population/population-composition/population-statistics/>

(11) Hálfðánarson ÓÓ, Pottegård A, Björnsson ES, Lund SH, Ogmundsdóttir MH, Steingrímsson E, Ogmundsdóttir HM, Zoega H. Proton-pump inhibitors among adults: a nationwide drug-utilization study. *Therap Adv Gastroenterol.* 2018 May 30;11:1756284818777943.

(12) <https://www.socialstyrelsen.se/en/statistics-and-data/statistics/statistical-databases/>

Reviewer: 3

Dr. Guido Knapp, Technische Universität Dortmund

Comments to the Author:

The authors used a retrospective cohort study to investigate the risk of colorectal cancer associated with long-term use of proton pump inhibitors.

The section on statistical analyses clearly describes the methods used. Maybe the authors should add a sentence or two about the propensity score adjustment. No information about the realization is given here.

>>Reply:

Thank you for your careful review and constructive feedback on our manuscript. We appreciate your thorough evaluation and have made the following revisions in response to your comments:

We have included a brief sentence detailing the logistic regression of the propensity score adjustment in the Statistical analyses section to enhance clarity: "Propensity score adjustment between maintenance PPI users and H2RA users was performed in the model to better control for confounders. Propensity scores were estimated using multivariable logistic regression, including age, sex, and indications."

Table 3: The lower forest plot is too high (between the lines). And most likely shifted to the righthand side, for instance, last confidence interval does not include one. Please correct!

>>Reply:

The forest plots in Table 3 have been removed to decrease disturbance and we have verified that the confidence intervals are correctly positioned.

Results 3. Common indications for maintenance PPI use: Confidence interval for dyspepsia in the text is not correct.

>>Reply:

Thank you for pointing out the discrepancy in the confidence interval for dyspepsia in the text. We have reviewed and corrected the confidence interval in the revised manuscript.

Table 4: Please check again the position of the forest plots! It is unclear how many patients have been analyzed for the Poisson regression model. Please add the information!

>>Reply:

We appreciate your feedback on Table 4. The forest plots have been removed for better clarity. Now we have emphasised the numbers in the Methods section: "The number of patients analysed for the Poisson regression model equals the summarized number of maintenance PPI users and maintenance H2RA users."

Thanks again for your correspondence and guidance. We look forward to any further suggestions you may have.

Sincerely,
Qing Liu, MD PhD
Nele Brusselaers, MD MSc PhD

Department of Microbiology, Tumour and Cell biology
Karolinska Institutet, Stockholm, Sweden

VERSION 2 – REVIEW

REVIEWER	Dai, Haibin Zhejiang University School of Medicine Second Affiliated Hospital, Pharmacy
REVIEW RETURNED	24-Apr-2024

GENERAL COMMENTS	1. Page 7, Line 41-45: "The information on the incidence of CRA and the demographic characteristics of the total population can be downloaded from the National Board of Health and Welfare's statistical database (summarised in the Supplementary Table S2)": Table S2 shows the number of CRA patients in Sweden and those subgroups with specific demographic characteristics. However, it does not show Sweden's total population and the corresponding subgroups' population. Similarly, there is no incidence of CRA in general and across subgroups. Please add the above data. 2. Page 9, Table 2: Please consider adding the cumulative CRA incidence and expected CRA patient number of maintenance PPI users and subgroups in Table 2.
---

REVIEWER	Knapp, Guido Technische Universität Dortmund
REVIEW RETURNED	22-Apr-2024

GENERAL COMMENTS	No further comments! Thank you!
---------------------------------

VERSION 2 – AUTHOR RESPONSE

Reviewer: 3

Dr. Guido Knapp, Technische Universität Dortmund

Comments to the Author:

No further comments! Thank you!

>>Reply: Thank you for your kind reply.

Reviewer: 2

Dr. Haibin Dai, Zhejiang University School of Medicine Second Affiliated Hospital

Comments to the Author:

1. Page 7, Line 41-45: "The information on the incidence of CRA and the demographic characteristics of the total population can be downloaded from the National Board of Health and Welfare's statistical database (summarised in the Supplementary Table S2)": Table S2 shows the number of CRA patients in Sweden and those subgroups with specific demographic characteristics. However, it does not show Sweden's total population and the corresponding subgroups' population. Similarly, there is no incidence of CRA in general and across subgroups. Please add the above data.

>>Reply: Thank you for pointing this out. First, we need to clarify that we don't obtain the number of Swedish total population when calculating the standardised incidence ratios (SIRs) but extract the observed CRA patient number of Swedish total population from our national registries, and compare this with the expected rates. We have clarified this in the methods as follows: " We calculated standardised incidence ratios (SIRs) with 95% confidence intervals (CIs) to assess the association between maintenance PPI use and CRA. Standardisation was performed according to Breslow and Day's method, and the allocation of person-years followed Clayton's algorithm (1). SIRs were obtained by calculating the ratio of observed CRA incidence rate of maintenance PPI users compared to the expected incidence rate derived from the incidence among the Swedish total population of the same age, sex and calendar period."

Second, we also added the incidence of the Swedish total population retrieved from the National Board of Health and Welfare in the methods as follows: " The information on the incidence of CRA (summarised in the Supplementary Table S2) and the demographic characteristics of the total population can be downloaded from the National Board of Health and Welfare's statistical database (2). During the study period (2005-2012), the overall incidence of CRA in men ranged between 81.8-89.9/100,000 individuals and between 74.8-79.7/100,000 individuals in women (presented in Supplementary Figure 2)." Please see the Supplementary Figure 2 as follows:

Figure S2. Charts showing the crude incidence of the Swedish total population. Data were retrieved from the National Board of Health and Welfare, Statistical Database.

(a) Crude incidence of new colorectal adenocarcinoma cases (over 20 years old) in Sweden (2012), by sex

(b) Crude incidence of new colorectal adenocarcinoma cases in Sweden (2006-2012), by sex

2. Page 9, Table 2: Please consider adding the cumulative CRA incidence and expected CRA patient number of maintenance PPI users and subgroups in Table 2.

>>Reply: Thank you for your comment, which we have discussed among the authors. Cumulative incidence may not be appropriate for this study, as we have an open cohort, meaning people can enter the cohort later (when they turn 18) and were also included at different ages. Yet we reported the observed CRA numbers and SIRs of maintenance PPI users and subgroups in each stratum of Table 2-4. As we mentioned before, we only retrieved the expected incidence rate from the Swedish total population. We may not report the expected CRA patient number of maintenance PPI users in Table 2. Instead, we reported the expected CRA number of the Swedish general population in Supplementary Table 2.

Reference:

(1) Machin D, Breslow NE, Day NE. Statistical Methods in Cancer Research, Vol. II: The Design and Analysis of Cohort Studies. 1990 Dec;46(4):1243.

(2) Statistics database - National Board of Health and Welfare [Internet]. Available from: <https://www.socialstyrelsen.se/statistik-och-data/statistik/statistikdatabasen/>

Reviewer: 3

Competing interests of Reviewer: None!

Reviewer: 2

Competing interests of Reviewer: None.

Overall, thanks again for the valuable review work and your suggestions.